# Tactile emoticons: Conveying social emotions and intentions with manual and robotic tactile feedback during social media communications

**Alkistis Saramandi**[1‡]*, **Yee Ki Au**[1‡], **Athanasios Koukoutsakis**[1], **Caroline Yan Zheng**[2,3], **Adrian Godwin**[4], **Nadia Bianchi-Berthouze**[5], **Carey Jewitt**[6], **Paul M. Jenkinson**[1,7], **Aikaterini Fotopoulou**[1]

1 Department of Clinical, Educational and Health Psychology, University College London, London, United Kingdom, 2 Royal College of Art, London, United Kingdom, 3 KTH Royal Institute of Technology, Stockholm, Sweden, 4 Independent Researcher, United Kingdom, 5 UCL Interaction Centre, University College London, London, United Kingdom, 6 UCL Knowledge Lab, Culture Communication and Media, University College London, London, United Kingdom, 7 Faculty of Psychology, Counselling and Psychotherapy, The Cairnmillar Institute, Melbourne, Australia

‡ AS and YKA have shared the first authorship on this work.

* alkistis.saramandi.15@ucl.ac.uk

**Data Availability Statement:** All our data and analysis code are available on github at: https://github.com/katlaboratory/Emoticon.

## Abstract

Touch offers important non-verbal possibilities for socioaffective communication. Yet most digital communications lack capabilities regarding exchanging affective tactile messages (tactile emoticons). Additionally, previous studies on tactile emoticons have not capitalised on knowledge about the affective effects of certain mechanoreceptors in the human skin, e.g., the C-Tactile (CT) system. Here, we examined whether gentle manual stroking delivered in velocities known to optimally activate the CT system (defined as 'tactile emoticons'), during lab-simulated social media communications could convey increased feelings of social support and other prosocial intentions compared to (1) either stroking touch at CT sub-optimal velocities, or (2) standard visual emoticons. Participants (N = 36) felt more social intent with CT-optimal compared to sub-optimal velocities, or visual emoticons. In a second, preregistered study (N = 52), we investigated whether combining visual emoticons with tactile emoticons, this time delivered at CT-optimal velocities by a soft robotic device, could enhance the perception of prosocial intentions and affect participants' physiological measures (e.g., skin conductance rate) in comparison to visual emoticons alone. Visuotactile emoticons conveyed more social intent overall and in anxious participants affected physiological measures more than visual emoticons. The results suggest that emotional social media communications can be meaningfully enhanced by tactile emoticons.

## Introduction

Social touch is key to human emotional communication, yet touch is absent from our burgeoning digital communications (e.g., chatting apps). Social media are a motivator for

**Funding:** This work was undertaken with funding from the UCL Social Science+ award (to CJ, NB and AF; https://www.ucl.ac.uk/research/domains/collaborative-social-science/research/social-science-plus), and in part with funding from the European Research Council (ERC) Starting Grant ERC-2012-STG GA13755 (to AF; https://erc.europa.eu/apply-grant/starting-grant), and the European Union's Horizon 2020 research and innovation programme under grant agreement No. 818070, for the Consolidator Award METABODY (to AF; https://erc.europa.eu/apply-grant/consolidator-grant). The funders had no role in study design, data collection and analysis, decision to publish, or preparation of the manuscript.

**Competing interests:** The authors have declared that no competing interests exist.

information sharing, fulfilment of social connection, and relationship maintenance (e.g., [1–3]). However, intense social media consumption (e.g., endless platform scrolling) may also hinder social connections, induce phubbing (the habit of favouring a mobile device over a physically present person) and negatively influence individual wellbeing (e.g., [4–8]; for reviews on social media, (dis)connection and mental health see [9, 10]). Crucially, typical social media platform communications mostly rely on vision and audition, both in (non-)verbal content sharing (e.g., texts) and feedback (comments and emojis). While we may be touching our phones during digital communications, such communications have a sensory restriction as they lack the communicative tactile signals which may be used in face-to-face, social interactions. However, to our knowledge, the effects of touch absence during digital communications have not been investigated.

Yet numerous studies have shown that social touch can convey meaning (e.g., love, support; [11–13]) and exert effects on others' emotions (e.g., soothe and buffer their (social) stress; [14, 15]) and behaviour (e.g., touch can increase restaurant tipping; [16]). For example, in a set of systematic, experimental studies, Hertenstein et al. [11, 17] demonstrated that healthy participants successfully read distinct emotions delivered through touch on their body, such as sympathy via stroking and patting. Moreover, others showed that familiar senders can convey messages of love, sadness, and gratitude, among others, by touching the receiver's forearm with intuitively suitable gestures for each specific message [13]. Furthermore, converted core features of 'tactile messages' to 'standardised' touch profiles were accurately decoded by participants even when provided by strangers. Finally, even simple caresses by strangers, delivered at the right velocity, can reliably communicate intimacy and social support, even without supplementary visual, or auditory clues [12]. Stroking, gentle touch delivered at relatively slow velocities (1-10cm/sec) has been associated with the activation of a particular afferent system, the C-tactile (CT) system. Accumulating evidence from a similar afferent system in mammals, and human microneurography, neuroimaging, neuromodulation, lesion and behavioural studies suggests that the activation of CT-tactile fibers at the periphery may be associated with increased likelihod of felt tactile pleasure ([18–22]; reviewed in [23]), and prosocial communication and effects ([12, 15]; reviewed by [24–27]). This kind of touch is spontaneously used by adult partners [28], and mothers to spontaneously stroke infants [29], with the latter capable to distinguish this touch from other kinds of touch, e.g., non-affective touch [30].

Slow, stroking touch, akin to hugs or hand-holding, communicates emotions, influences behaviour, and also has regulatory effects on others' physiology [31]. From birth, our caregivers use touch to regulate physiological, bodily states such as pain and hunger by stroking or holding, and feeding us [32], shaping our lifelong affect regulation (see [27] for a theoretical review). This social touch, and particularly CT touch, reduces negative physiological and affective states such as the pain of social rejection ([15]; for review see [27, 33]), physical and social pain [34–41], stress (for reviews see [25, 42]), pain during pain anticipation [43] and pain experience [15, 39–41, 44]. Apart from regulating negative affect, social touch exerts positive effects on wellbeing ([45–47]; for a review see [48]), partially through promoting social affiliation [49], and its perception [50]. At a neurobiological level, bonding may be supported through increased endogenous μ-opioid activity and oxytocin release associated with affective touch (e.g., [51, 52], for reviews see [53, 54]).

Given these aforementioned benefits of touch, several technologies have been used to design and develop devices and robots, mainly at 'prototype' level, that would allow tactile messages and experiences to be exchanged digitally by physically distanced people (mediated touch; for reviews and overview of haptic technologies see [26, 55–57]). For instance, Kissenger [58]–an interactive, kiss transmission device–was developed to satisfy needs for intimacy and social connection between two remotely connected people. A prototype with similar aims

was developed for parents and children [59]; the Huggy Pajama, a hug input device, simulates the feeling of receiving a hug. The benefits of such mediated, social touch technologies include increased feelings of proximity, and a positive affective experience between the touch giver and receiver (for more examples and benefits see [60–66]).

Yet, the benefits of mediated, affective touch remain extensively underexplored in social media communications. The growing prominence of affective computing has created the discussion of various state-of-the-art applications and the review of the benefits of touch (for reviews see [67, 68]). Preliminary survey results indicate a preference for tactile-enhanced voice communication [69] and a web-application prototype (known as Haptic-Emoticon) has been developed to enrich basic social media communication [70, 71]. However, these technologies remain tested at an initial prototype, and feasibility level, and the implementation of tactile signals is typically not informed by the neurobiology of touch.

Conversely to these nascent haptic or tactile emoticon developments, most research on 'emoticon-based' communications has focussed on text-embedded visual emoticons (e.g., facial expression icons) to express certain emotions. Emoticons often strengthen a message when touch, facial expressions, vocal tone and eye contact are absent [72]. The usage patterns of emoticons align with that of facial behaviours–they are more likely to express humour, in positive contexts, friendly interactions, and in situations of high emotional expressivity [73–75], and their meaning and use are context-dependent. For example, secondary school students responded to short internet chats by picking an available emoticon (e.g., smile, wink, etc.), or by using emoticon-text combinations [76]. Participants generally used more emoticons for socio-emotional chats (compared to task-oriented ones), and favoured positive emoticons for positive contexts and more negative emoticons for negative contexts [76]. Yet, no study has examined the corresponding effects and efficacy of tactile emoticons compared to visual emoticons. Given the aforementioned role of tactile interactions, especially slow stroking at CT-optimal velocities for expressing prosocial emotions, we tested the effects of mediated-stroking (i.e., tactile emoticons) on the perception of social approval and support during social media communications in comparison to (Experiment 1) or in addition to (Experiment 2) typical visual emoticons.

Specifically, Experiment 1 simulated a social media communication platform during which healthy adults imagined they were reading their own typed posts (of either positive or negative emotional valence), while a confederate delivered tactile (using a brush) or visual feedback that communicated high support (e.g. stroking touch at CT-optimal velocities as a tactile emoticon or a red heart or thumbs up as a visual emoticon), or low support (e.g., stroking touch at CT-suboptimal velocities or a neutral blue heart as a visual emoticon). We predicted that participants would perceive greater social support and approval when receiving tactile feedback in comparison to visual feedback, irrespective of the level of support or the posts' valence. In a follow-up, preregistered, confirmatory study (Experiment 2) we used a similar setup but delivered affective 'stroking touch' with a validated, soft robotic sleeve (S-CAT; [77]), which elicits affective touch at CT optimal speeds (see Methods for details on the properties of the S-CAT). We investigated whether combining the above affective tactile emoticons with standard visual emotions would lead to greater perception of social approval and support than visual emoticons alone. We also predicted to see a greater downregulation of physiological measures (e.g., heart rate and skin conductance rate) following visuotactile vs visual feedback, given associations of touch with lower blood pressure, heart rate, and decreased anxiety [78–82]. Additionally, we looked at various contributing factors such as identification with and relevance to the posts, and feelings of trust and safety while wearing the S-CAT to explore whether they influenced participants' perception of approval and support. In both experiments, we also examined the moderating role of certain key individual differences such as social attitudes to touch.

## Methods: Experiment 1

### Participants

A power analysis on G*Power [83], determined that N = 36 participants should be recruited to achieve the smallest effect size of interest [84] (see S1 File for details). A total of N = 39 participants with no prior history of psychiatric, neurological, and/or dermatological conditions were recruited from the University College London (UCL) research participant database (SONA; testing took place between 11 June 2019 and 30 September 2019), but two participants were removed from the dataset due to noncompliance with instructions and one due to software failure, respectively; final sample N = 36 (50% male; $M_{Age} = 25.08$, $SD_{Age} = 3.83$; Table 1). The sample demographic information is consistent of the typical UCL student population. The UCL Department of Research Ethics Committee granted ethical approval for this experiment, all participants provided written informed consent and were rewarded a fixed sum of £10 for their time.

### Design

We used a 2 (valence: positive vs. negative) x 2 (feedback mode: tactile vs. visual) x 2 (feedback support level: high vs. low) within-subjects design. Valence was manipulated by sentence stimuli (64 in total; 32 per valence), presented as Twitter-like posts. Feedback mode and support level were manipulated by the provision of tactile and visual emoticons, each at two levels of support, as follows: Tactile emoticons consisted of brief brush strokes of the skin at either CT-optimal velocities (3cm/sec or 6cm/sec), expected to convey support most clearly (feedback support level: high), or CT -suboptimal velocity (0.3cm/sec), expected to convey social support less clearly (feedback support level: low). Visual emoticons consisted of emoticon images with specific, socially supportive meanings (high support: red heart, or thumbs up), or less specific meanings (low support: neutral blue heart), expected to convey support less clearly. The different types of high tactile and visual feedback support level stimuli were used to reduce psychological habituation but emoticon type was not used as a confounding factor in our main analysis (i.e., we did not compare between ratings following 3cm/sec vs 6cm/sec tactile emoticons and red heart vs. thumbs up visual emoticons). In total, we had 8 conditions, with 8 trials in each, corresponding to the different sentences (hereafter referred to as posts; Fig 1). Feedback mode and support level were counterbalanced, and presentation order was pseudorandomised according to sentence valence and context. Thus, our main independent variables (IVs) were valence, feedback mode and support level.

**Table 1. Demographics of Experiment 1 participants.**

|  |  | N | % |
|---|---|---|---|
| GENDER | Female | 18 | 50.00 |
|  | Male | 18 | 50.00 |
| SEXUAL ORIENTATION | Heterosexual | 28 | 77.78 |
|  | Homosexual | 3 | 8.33 |
|  | Bisexual | 2 | 5.56 |
|  | Other | 1 | 2.78 |
|  | Prefer not to say | 2 | 5.56 |
| ETHNICITY | White | 14 | 38.89 |
|  | Asian | 15 | 41.67 |
|  | Other* | 7 | 19.44 |

Note. Other* refers to one of the following: Middle Eastern, African American, Latino, and Mixed Race.

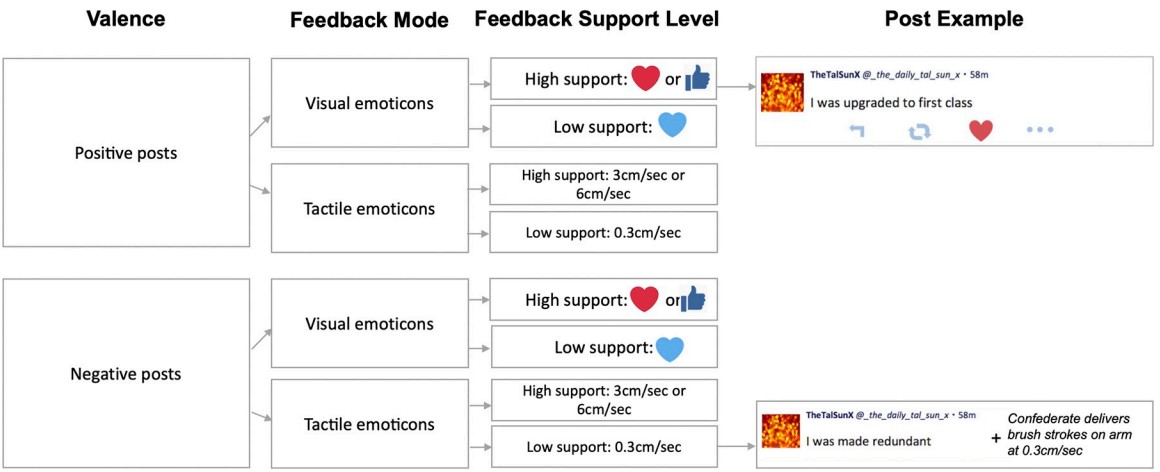

**Fig 1. A visual illustration of the experimental design and posts of Experiment 1.**

The main dependent variable (DV) was the amount of 'social approval and support' participants rated after each feedback trial (hereafter referred to as 'social intent') on a 0 (*no approval and support at all*) to 100 (*extreme approval and support*) scale.

In Experiment 1 there were three main measures: Valence (i.e., the Valence of the posts' context: Positive vs. Negative), Feedback Mode (Visual vs. Tactile emoticons) and Feedback Support Level (High vs. Low support). Specifically, in the positive valence half of the posts were paired with predetermined Visual feedback and the other half were paired with predetermined Tactile Feedback. The same was repeated for the posts with a negative valence. Further, within each Feedback Mode, the feedback participants received on half of the posts was of high Feedback Support Level (in the Visual feedback mode: red heart, and thumbs up emoticon; and, in the Tactile Feedback Mode: touch at 3cm/sec, and 6cm/sec), while in the other half they received low Feedback Support Level (in the Visual Feedback Mode: neutral blue heart; in the Tactile Feedback Mode: touch at 0.3cm/sec). This design led to 8 possible combinations and 8 posts per combination (i.e., 64 posts in total). The two post examples are examples of the post format used in the main task, showing what participants saw when getting feedback from the confederate. Participants were told in the beginning of the experiment that 'TheTalSunX' would be their username and '@_the_daily_tal_sun_x' was the respective username of the platform from which they would be posting and getting Visual or Tactile feedback from. Every post displayed the username and handle on the top, as well as a profile picture (unbeknownst to them, the same for all participants) on the top left corner of the post, which consisted of a neutral image (bright red and orange dots). The timestamp on the top right corner of the post indicated how long ago the participant had posted on their profile (here, 58 minutes ago; all posts across all participants had the same timestamp). Below these standardised features all participants could see the sentence they posted (first, without the feedback, and then with the feedback). The example at the top right corner of the figure is a post of individualistic context, and positive valence ("I was upgraded to first class") which received visual feedback from the confederate (here, high support as indicated by the 'red heart' emoticon). The example at the bottom right corner of the figure is a post of individualistic context and negative valence, which then received tactile feedback from the confederate (here, low support via brush strokes at 0.3cm/sec). The icons below the sentence would only appear when the participant was receiving visual feedback, and they are visually similar (although inactive here) to what users of actual social media platforms typically see when interacting which each other. The features,

starting from the left are 'reply', 'reshare', 'react', 'more', the latter offering more functionalities.

**The creation of social media posts as experimental sentence stimuli.** Our verbal stimuli were created based on a pilot study with a separate sample (N = 13, N = 8 female; $M_{age}$ = 23.15, $SD_{age}$ = 4.06). Specifically, we first reviewed the structure and content of tweets posted on popular *Twitter* (recently rebranded as X) accounts to extract the typical format–consisting of a profile picture, username, timestamp, and brief sentences (max. 280 characters), with an average of 70 characters per post. Then, we constructed individual brief social media posts according to the aforementioned 'tweet' format, which in Experiment 1, as well as in Experiment 2 (see below) were displayed (in the middle of a PC screen in front of participants) one at a time to create a *Twitter*-like experience (Fig 1).

To construct the content of these sentences, we first created a pool of 48 brief sentences that describe various possible everyday events that could be shared on social media (e.g., *I got a promotion*; *I watched a horrible movie*). These constructed sentences may be accompanied by various emotions—ranging from basic emotions such as joy, sadness or, anger [85, 86] to more complex, secondary emotions such as love, shame and disgust [87] and could therefore be subject to supportive, or approving feedback in the main experiment (see below). Half of the sentences were of positive valence and the other half were negative (e.g. *Happy and relieved I passed my French exam* vs. *Sad that my dinner plans with friends next week got cancelled*). Additionally, for variability and some ecological validity, sentences were written in a way to reflect both social and individual contexts (n = 24 social; n = 24 individual; e.g., *We are going on holidays with my friends* vs. *I made a serious mistake at work*). Then, participants rated the valence and context of the sentences (on a scale of -10 (*extremely negative*) to +10 (*extremely positive*), and -10 (*extremely individualistic*) to +10 (*extremely social*), respectively. Participants successfully distinguished the different valences $t(46)$ = 23.186, $p < .001$ ($M_{Negative}$ = -5.33, $SD_{Negative}$ = 1.90; $M_{Positive}$ = 6.05, $SD_{Positive}$ = 1.47) and contexts $U$ = 31.50, $p < .001$ ($M_{Individualistic}$ = -5.27, $SD_{Individualistic}$ = 2.07; $M_{Social}$ = 2.90, $SD_{Social}$ = 3.94). Participants also commented that the sentences were quick to read and process but too uniform, suggesting to add greater content variability. Hence, we added another 16 sentences of similar format and valence/context categories, yielding a total of 64 sentence-stimuli for Experiment 1, all designed in the aforementioned format (S1 Table in S1 File).

**Creation of affective visual feedback stimuli with visual emoticons.** In the main experiment, the aforementioned sentences were displayed together with either visual or tactile feedback of different specificity (feedback support level) and valence. To manipulate visual feedback, three widely used visual emoticons (high support: red heart, or thumbs up; or of less specific meaning, low support: neutral blue heart) were displayed beneath the sentence posts, as shown in Fig 1. The high and low support emojis were chosen following the emojis social media users can typically use to engage with a post (e.g., a red heart on X, and a blue thumbs up, or a red heart, among others to express "like" and "love", respectively, on Facebook). To examine whether this separation of our visual emoticons did indeed affect participants' perceived social intent, feedback support level was added as a IV in our main analyses.

**Affective tactile feedback with tactile emoticons.** A confederate (see more below) then delivered different types of feedback (tactile via brush strokes on the participant's forearm, or visual feedback via emoticons underneath the post) to the participant. The feedback was used to convey feelings of approval and support from the sender (confederate) to the receiver (participant), based on previous work showing that touch at optimal speeds may reliably communicate various emotions [11, 12, 17], including emotions typically described as 'primary' (e.g., joy, sadness) and secondary (e.g., admiration, contempt).

During the experiment, half of the posts received only tactile emotions (touch via brush strokes in three velocities; 0.3, 3, and 6 cm/s), manipulated as follows. Prior to the main task, the experimenter drew two 9x4cm rectangles on the participant's left dorsal forearm area with a washable marker to indicate where affective touch should be applied (using a soft cosmetic brush; Natural Hair Blush Brush, N.7, The Boots Company, 1.5cm lateral width). To minimise habituation and to prevent CT-afferent fatigue, affective touch was alternated between the two marked skin areas. To control for the pressure applied, the lateral spread of the brush bristles during affective touch delivery always remained within the marked rectangles and touch was delivered on the forearm in a proximal-to-distal direction. The confederate was trained to deliver touch for a duration of 3 seconds per trial—i.e., from first contact to cessation of contact after the appropriate number of strokes the duration was 3 seconds (e.g., 3cm/sec was 1 continuous stroke for 3 seconds, given the rectangle's length of 9cm). Overall habituation concerns were also addressed and minimised by fully randomising the (visual and tactile) feedback order within and between participants.

## Procedure

**Role allocation: 'Feedback receiver' vs a confederate 'feedback giver' introduced as a peer participant.**   Two 'participants' (actually one participant and one confederate; see below) were invited into the experimental room and told that they would be randomly assigned to different roles; one would have the role of the social media post writer and 'feedback receiver', and the other would be the 'feedback giver'. Participants were introduced to the 'other participant' taking part in the experiment as though they were a peer volunteer, whereas they were actually another experimenter acting as a confederate. It was stressed to participants (always feedback receivers) that the feedback giver could not see their ratings (i.e., perceived social intent) so that the (visual or tactile) feedback delivered by the feedback giver would only be influenced by the content of the displayed posts. In reality, the confederate was always the feedback giver and followed the script instructions (which the participant could not see) to deliver relevant feedback—either visual emoticons delivered by pressing the keyboard of a laptop in front of them, or by brush stroking. The two were asked to not interact verbally throughout the experiment unless instructed to do so.

**Set-up.**   The experiment was programmed in Psychopy v3.2 [88] and ran on a 13" Dell laptop, with 800 x 600 pixels, display size 100%. Participants sat at a table in front of the experiment laptop they would use during the task. The confederate sat directly opposite the participant at the table and observed the participant's viewed posts through a display monitor synced to the experiment laptop. The participant and confederate were only able to see their own display screen and keyboard. Brief introductions to social media and their features were given to ensure familiarity with the concept of *Twitter*, tweets/posts and 'likes'. Participants were then asked to take the perspective of a social media user and imagine that the constructed posts had been posted by them on their own profile, and they were informed that they would receive feedback by the 'other participant' in the form of either visual or tactile emoticons, and they had to rate this feedback. The 64 posts were displayed to the participant one-by-one, and after 10 seconds, the confederate delivered feedback: either visual feedback (emoticons) or tactile feedback (affective touch) to the participant for 3 seconds. After each trial, participants were immediately asked "*How much social approval and support did you feel*?" and reported their rating on a scale of 0 *(not at all)* to 100 *(extremely)*. Finally, participants completed the psychometric questionnaires and were fully debriefed. A manipulation check question revealed that no participant guessed this experimental deception in advance.

## Data analysis

Analyses for both experiments were conducted with R (R, Boston, MA). A preliminary analysis to check whether the variability in sentence context (i.e. individual vs. social) had any unexpected effects on social intent did not yield significant differences between individual and social sentences and explained only a very small amount of variance in social intent scores ($\beta$ = 0.897, $SE$ = 1.195, $t$ = 0.75, $p$ = 0.453, $R^2_{adj}$ = -0.0002). Hence, as planned, context was not included as a main effect or variable in subsequent analyses.

We conducted multilevel modelling (MLM) for each prediction. For Hypothesis 1 we examined whether feedback mode (IV) influenced participants' perceived social intent (DV). To examine whether feedback support level and valence influenced participants' perception of social intent we ran two separate analyses with feedback support level and valence as the respective IVs. We also examined the two-way interactions between feedback mode and feedback support level, and feedback mode and valence. Finally, we ran a three-way interaction between feedback mode, support level and valence (IVs) with social intent as our DV. In all analyses, participant ID was used as our random effect. Our secondary and exploratory analyses are presented in the S1 File.

## Results: Experiment 1

### Hypothesis 1: Participants preferred tactile feedback over visual feedback

We found that overall participants perceived significantly greater social intent after receiving tactile vs visual feedback ($\beta$ = 12.48, SE = 1.00, $t$(12.52), 95% CI = 10.52–12.43, $p$<0.001, ICC = 0.29, $R_{marginal}^2$ = 0.047, $R_{conditional}^2$ = 0.322; Fig 2). Participants also reported greater perception of social intent following high vs low feedback support level ($\beta$ = 3.41, SE = 1.00, $t$(3.42), 95% CI = 1.46–5.36, $p$ = 0.001, ICC = 0.35, $R_{marginal}^2$ = 0.003, $R_{conditional}^2$ = 0.350). Yet, the two-way interaction between feedback mode and support level was not statistically significant ($\beta$ = -2.27, SE = 1.99, $t$(-1.41), 95% CI = -6.18–1.63, $p$ = 0.254, ICC = 0.28, $R_{marginal}^2$ = 0.051, $R_{conditional}^2$ =

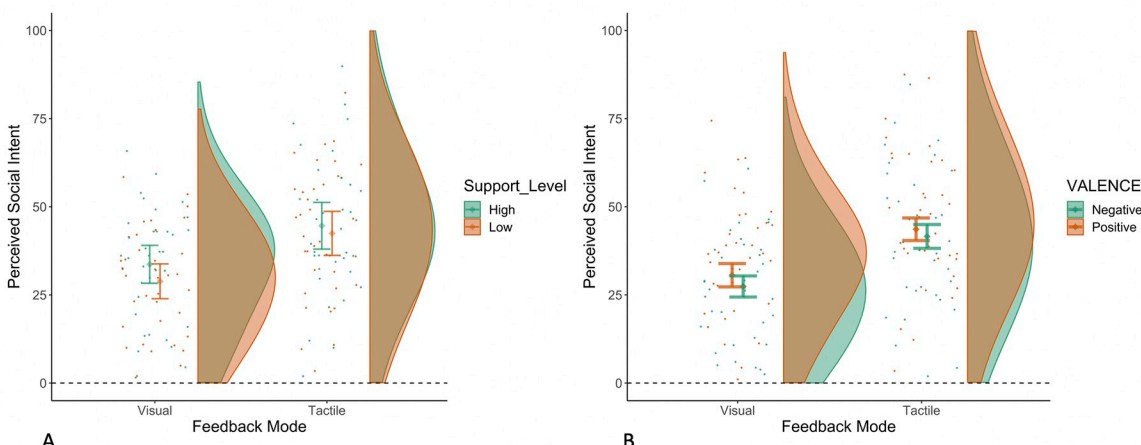

**Fig 2. Social intent scores following visual and tactile feedback.** Panel A shows participants' social intent scores following feedback at high and low support levels. Panel B shows participants' social intent scores following feedback (high support only) on positive and negative posts. The rainclouds represent the distribution of Social Intent when grouped by Feedback Mode and Support Level (in Panel A) and when grouped by Feedback Mode and Valence (in Panel B). The dots represent the average score per participant for each combination of Feedback Mode and Support Level (in Panel A) and Feedback Mode and Valence (in Panel B). Error bars indicate the 95% confidence interval (CI) around the mean (Mean of Social Intent +/- 95% CI). The dot on the error bar is the sample mean of Social Intent for the respective combination of Feedback Mode and Support Level (in Panel A) and Feedback Mode and Valence (in Panel B). The horizontal dashed line (at y = 0) indicates that the Social Intent scores could not be below 0.

0.320). Next, we also found a significant effect of valence: greater social intent scores were seen in the positive vs negative sentences ($\beta$ = 7.22, SE = 1.00, $t$(7.24), 95% CI = 5.26–9.17, $p$<0.001, ICC = 0.33, $R_{marginal}^2$ = 0.015, $R_{conditional}^2$ = 0.342) and there was non-significant trend on the two-way interaction between feedback mode and valence ($\beta$ = -3.45, SE = 1.99, $t$(-1.73), 95% CI = -7.35–0.45, $p$ = 0.083, ICC = 0.26, $R_{marginal}^2$ = 0.065, $R_{conditional}^2$ = 0.312). To follow-up on the aforementioned trend and better understand how feedback mode was perceived differently within each valence separately, we examined each level of the factor 'valence' separately and found that tactile feedback led to significantly greater levels of perceived social intent in comparison to visual feedback in both cases (Positive Valence: ($\beta$ = 10.85, SE = 1.45, $t$(7.48), 95% CI = 7.98–13.65, $p$<0.001, ICC = 0.28, $R_{marginal}^2$ = 0.034, $R_{conditional}^2$ = 0.308); Negative Valence: ($\beta$ = 14.11, SE = 1.30, $t$(10.87), 95% CI = 11.56–16.65, $p$<0.001, ICC = 0.32, $R_{marginal}^2$ = 0.066, $R_{conditional}^2$ = 0.367)), thus suggesting that there was no meaninful interaction between feedback mode and valence at least as tested in this experiment. Finally, the three-way interaction between feedback mode, feedback support level and valence was not significant ($\beta$ = -4.44, SE = 3.97, $t$ (-1.12), 95% CI = -12.21–3.33, $p$ = 0.263, ICC = 0.26, $R_{marginal}^2$ = 0.075, $R_{conditional}^2$ = 0.317). In sum, tactile vs visual emoticons, high versus low support level and positive versus negative posts, lead to greater perceived social intent, but these factors did not interact between them.

## Methods: Experiment 2

### Participants

Based on the effect size ($f$ = 0.203) from the trend between valence and feedback mode in Experiment 1, the minimum number of participants and 95% power for Experiment 2 was N = 54 (see S1 File for details). Fifty-six participants were initially recruited and participated in the experiment between the 29th of November 2021 and 7th of February 2022 but N = 4 were excluded due to a software error (order of blocks was not congruent with the pre-determined feedback mode). The final sample consisted of N = 52 participants ($M_{Age}$ = 22.04, $SD_{Age}$ = 4.44; N = 37 female; Table 2) and as in Experiment 1, was also characteristic of the typical UCL student population. An additional N = 5 and N = 8 participants were excluded from the HR and SCR analyses, respectively, due to a technical error and lack of sufficient physiological data collection throughout the session. However, a post-hoc power calculation showed that our power was 90% even for our smallest sample analysis (N = 44; for the analyses with skin

**Table 2. Demographics of Experiment 2 participants.**

|  |  | N | % |
|---|---|---|---|
| GENDER | Female | 38 | 73.08 |
|  | Male | 13 | 25.00 |
|  | Other | 1 | 1.92 |
| SEXUAL ORIENTATION | Heterosexual | 34 | 65.38 |
|  | Homosexual | 0 | 0.00 |
|  | Bisexual | 5 | 9.62 |
|  | Other | 9 | 17.31 |
|  | Not sure | 3 | 5.77 |
|  | Prefer not to say | 1 | 1.92 |
| ETHNICITY | White | 21 | 40.38 |
|  | Asian | 26 | 50.00 |
|  | Other | 5 | 9.62 |

Note. Other* refers to one of the following Middle Eastern, African American, and Mixed Race.

conductance rate (SCR); see S1 File). The eligibility criteria, recruitment sites, consent, compensation, and ethics were the same as those of Experiment 1.

## Main aims, design and data analysis

Based on the results of Experiment 1, where we found that overall participants preferred tactile feedback over visual feedback, in this preregistered follow up study (Experiment 2; see preregistration here: https://osf.io/f9sjv) we examined whether visual feedback (extensively used in current social media communications) combined with affective touch (here delivered via a wearable robotic sleeve) would elicit greater levels of perceived social intent in comparison to visual feedback alone. Moreover, in Experiment 1 we found that a red heart was not always conveying approval and support in response to negative scenarios and valence had an effect in how people interpreted our main measure, namely the question "*How much approval and support did you feel*?". Thus, in this Experiment 2, visual feedback, and social intent question were adjusted per valence (e.g. 'thumb up' vs. 'thumb down' and social approval and validation vs social support and sympathy, respectively). Next, given that perception of social intent may be influenced by the extent to which participants managed to imagine the posts were their own, we assessed whether identification with and relevance to the post influenced perceived social intent. Additionally, receiving touch from a wearable sleeve may be considered a novel experience and thus we investigated how feelings of safety and trust regarding the wearable sleeve influenced participants' response to the tactile component of the visuotactile feedback. We also measured the effects of feedback mode on physiological measures, such as heart rate, and skin conductance rate. These physiological, exploratory analyses were secondary aims of the present paper and the results are briefly reported below and reported in detail in the S1 File. Finally, we controlled for the pleasantness elicited following affective touch delivered by the S-CAT vs. manual brush strokes to ensure that the S-CAT can elicit pleasantness ratings comparable to brush-stroke touch when delivered at CT-optimal velocities. The details and results of this analysis are reported in the S1 File.

We used a 2 (feedback mode: visual vs. visuotactile) x 2 (valence: positive vs. negative) within-groups design to test the effects of these factors on the perception of social feedback during social media communications. Sentence stimuli (48 in total) were presented as Facebook-like posts (see below) with content of either positive or negative valence (24 sentences per valence). Feedback mode was manipulated by the provision of visual-only versus visuotactile emoticons. The visual component was identical between the conditions but the tactile component of the visuotactile feedback was delivered via a wearable sleeve which can deliver affective touch at CT-optimal speeds (6cm/sec) and whose properties and effects have been described and validated in a separate study (S-CAT; [77]; see details below). To add variance to the kind of visual feedback participants received, we chose 6 emoticons per valence which are typically used to convey socially supportive meanings at different intensities. In the posts of positive valence, the visual emoticon was a simple smiley face or thumbs up (low intensity), a red heart or clapping hands (medium intensity), or a smiley face with the excited star-eyes or a party hat (high intensity). Similarly, for the posts of negative valence, the visual emoticon was a sad face or thumbs down (low intensity), a shocked or crying face (medium intensity), an angry face or a hand with a heart to express care (high intensity). Within each valence category, eight sentences were paired with likes from a 'low likes' range (20–25 likes), eight from a 'medium likes' range (40–50 likes), and eight from the 'high likes' range (80–100 likes). The different emoticons and like ranges and sentence focus (individual vs social; see also Experiment 1) were used for ecological validity and to avoid habituation, and as preregistered, the type of emoticon, sentence focus and 'likes range' were not included as covariates in any of our

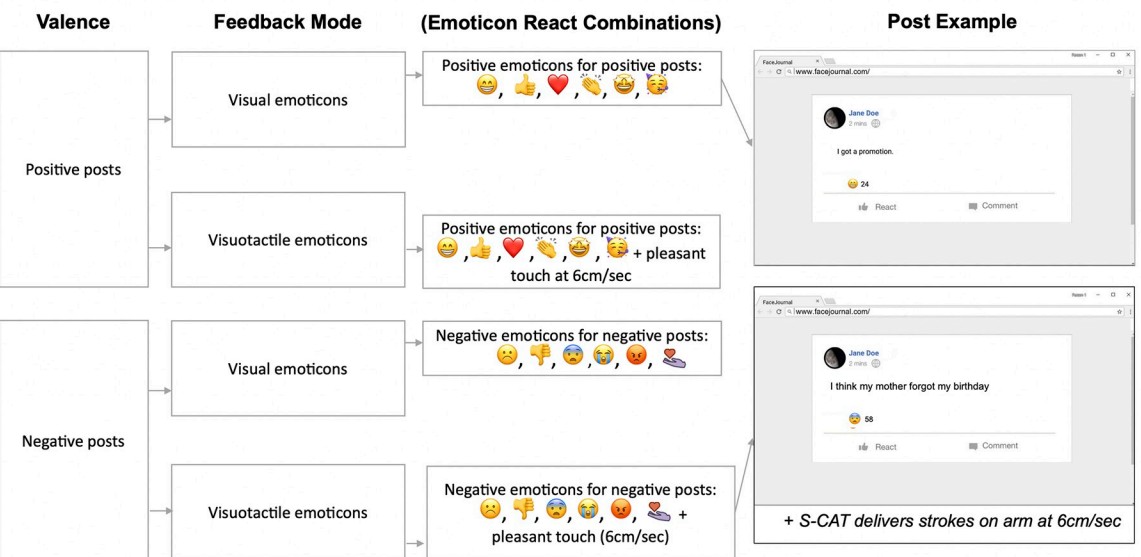

**Fig 3. A visual illustration of the experimental design and posts of Experiment 2.** In Experiment 2 there were two main measures: Valence (Positive vs. Negative) and Feedback Mode (Visual vs. Visuotactile emoticons). Participants received visual and visuotactile feedback on posts of positive and negative valence. There were 4 possible combinations (e.g., "Valence: Positive, Feedback Mode: Visuotactile Emoticons) and 8 posts per combination (i.e., 48 posts in total; see below). and within each Valence and Feedback Mode combination the number of and type of visual emoticon reacts was varied (low, medium, and high; see below). However, for additional variability within each of the 4 combinations we varied the type of visual emoticon react (2 per intensity: low, medium, high; see below), thus resulting in 2 posts per Valence, Feedback Mode and Emoticon Reach combination. We also varied the number of emoticon reacts (low, medium, and high; see below). The number and type of emoticon reacts were not taken into consideration in the Experiment 2 analyses. The post examples are examples of what a participant would see when logged onto their 'FaceJournal' profile and receiving visual feedback. The screen display was similar to that of actual social media platforms on a browser (i.e., with the name of the website and the link on the top left, as well as typical browser functionalities on the top right (e.g., 'bookmark'). Then, each post was individually shown like in this example. On each trial, participants' name of choice (like the 'Jane Doe' example shown here) and the given icon they had set for their profile (always a round abstract image as the one shown here) were shown at the top left corner of the post. Below their name, participants could see how many minutes ago they posted on their profile (here, 2 minutes ago; all posts across all participants had the same timestamp), with the 'connect' logo of the platform next to the timestamp (as indexed by the grey earth-like icon). In the example at the top right corner of the figure, a post of positive valence is shown and thus the visual feedback is a smiley visual emoticon (i.e., a low intensity visual emoticon) with seemingly 24 such reactions (i.e., a number of likes from the low range). To create a more realistic experience of FaceJournal interactions, below the other users' reactions participants could also see the 'React' and 'Comment' functionalities social media users also have below their own posts (should they wish to also react on their post and/or comment). The example at the bottom right corner of the figure is a post with the same design features as the top post, but with a sentence of negative valence and thus the emoticon react is a shocked face (here, medium intensity). The specific post was also paired with predetermined visuotactile feedback, thus the participant also received affective touch via the S-CAT at the same time as the visual emoticon appeared on their screen.

analyses. The sentences were divided into four blocks and block order was counterbalanced between participants based on the feedback mode, while post order within each block was counterbalanced based on valence (Fig 3). The feedback mode was predetermined for each post and the order of sentences within each block was randomised between participants.

As preregistered, for our first hypothesis, we examined whether feedback mode (IV) influenced participants' perceived social intent (DV), when using valence as a random effect, expecting visuotactile feedback to lead to higher levels than visual feedback alone (see above for justification of this hypothesis). We also examined the potential interaction between feedback mode and valence, and then ran two separate MLMs (one per valence; as preregistered) to examine the effect of feedback mode irrespective of the two-way interaction result.

For the second hypothesis we examined whether increased levels of identification with and relation to the posts (on a scale of 0 (*not at all*) to 100 (*extremely*)) would be predictive of increased levels of perceived social intent. We first ran a non-preregistered principal

component analysis (PCA) to orthogonalise our two IVs (i.e., identification and relevance), given the high correlation between our two measures. Then, as preregistered, we used social intent as the DV and examined the main effect of identification and relevance and then the interaction between feedback mode and identification, and feedback mode and relevance. Valence and participant ID were added as random effects. Although not initially preregistered, we also examined whether valence itself (IV) had an effect on participants' reported identification (DV).

For the third hypothesis we assessed whether higher amounts of perceived social intent in posts that received visuotactile feedback were predicted by participants' feelings of Safety and Trust (on a scale of 0 *not at all* to 100 *extremely*) when receiving touch. We used social intent as DV and ran analyses on the trials which only received visuotactile feedback, as obviously visual only trials did not involve any issues of touch 'safety' and 'trust'. Specifically, as preregistered, we examined the main effect of Safety, the interaction between Safety and valence, and finally the effect of Safety in each valence separately, irrespective of the two-way interaction effect on social intent, given the potential, aforementioned ambiguity emoticons may have had on the sentences of negative valence. We repeated these analyses using Trust as the IV instead of Safety, as preregistered. Participant ID and block were added as random effects.

For the fourth hypothesis, we examined the effect of feedback mode on three physiological measures, namely HR, SCR, and heart rate variability (HRV). We ran an MLM with baseline corrected HR, SCR, and HRV scores as our separate DVS, feedback mode as the IV and participant ID and block as random effects. We then examined the interaction between feedback mode and traits (rejection sensitivity and attachment anxiety). Details on physiological measure calculations are presented in the S1 File as the examination of feedback mode on physiological measures was a secondary aim of Experiment 2.

Where the variance of our random effects was equal to 0 (and Conditional $R^2$ output displayed as "N/A") the random effect was removed from the model. In case all random effects had to be removed, we conducted linear regressions instead (and explicitly report it).

## Materials & Set-up

**Selection of the social media posts as experimental sentence stimuli.** Our verbal stimuli were a selection from the sentences used in Experiment 1. In preregistered Experiment 2 we examined whether visuotactile feedback (i.e., tactile feedback in addition to visual feedback) would be associated with increased levels of perceived social intent as opposed to visual feedback only. Contrary to Experiment 1, in Experiment 2 we did not explore feedback optimality differences. Consequently, we reduced the number of sentences from 64 to 48. Additionally, given other time constraints (e.g., sentences in Experiment 2 were displayed for a longer duration in total), and feedback from Experiment 1 participants reporting that the numerous trials could become tiring we deemed that having fewer sentences overall would be appropriate. The sentences from Experiment 1 which received the highest average amount of perceived social intent were selected for Experiment 2 –we chose 24 positive and 24 negative sentences (S1 Table in S1 File). The sentences were converted into a Facebook-like post format (post font and size was Roboto 18) and were pre-uploaded on a simulated, virtual platform we created for Experiment 2, hereafter referred to as FaceJournal (see Fig 3).

**Creation of the FaceJournal platform.** Participants were introduced to our in-house, software interface called 'FaceJournal' at the beginning of the experiment and were told this was a new social media platform we were testing and they were invited to try out to give us information on what worked well and what not. FaceJournal, which was only created for these experimental purposes, would make this personalised experience of social media usage and

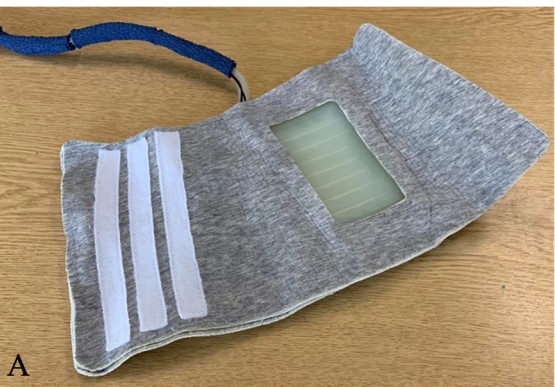
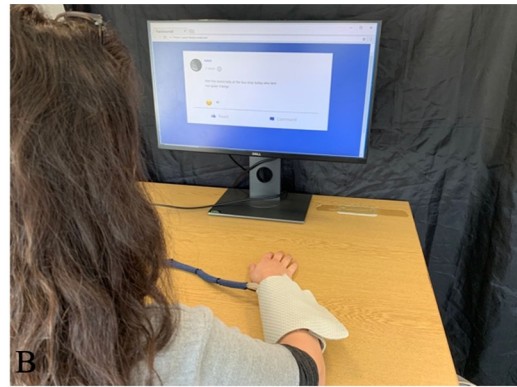

**Fig 4. The S-CAT and FaceJournal interface.** (A) Shows the inside materials of the S-CAT. This is the side that contacts with the skin, and the pneumatic actuators covered by the silicon layer on the outside (as shown in the image) were positioned on the participant's forearm, as it is where touch was being delivered from (for more see [77]). (B) Shows a participant during the session, wearing the S-CAT on their left forearm, and looking at their post on the FaceJournal platform, while receiving visuotactile feedback via the S-CAT and the visual emoticon on the screen. The participant is sitting behind the makeshift wall and cannot see the experimenter sitting behind it, delivering the tactile feedback. (Note. The participant has consented).

communication more realistic. FaceJournal was created on C# and ran on the same laptop and display settings as Experiment 1. The laptop was connected to a monitor to allow the experimenter to see the participant's screen display (unbeknownst and not visible to the participant). This allowed the experimenter to deliver tactile feedback via the S-CAT according to the instructions on the appropriate trials.

**Tactile feedback equipment and set-up.** Tactile feedback was delivered via the S-CAT (Fig 4), a silicone, pneumatic, soft haptic device which is able to imitate human touch via velocity, force and temperature. The device wraps around the participant's forearm (Fig 4 below), and the actuator design (with dimensions of 9cm x 4cm) which is covered with silicon on the inside is where participants feel the touch coming from. Specifically, the S-CAT has an array of pneumatic actuators which are arranged in a way to minimise spacing and is programmed to produce a tracing effect to simulate a caress-like touch gesture when actuated in a sequence [79]. In other words, the eight air cells of the device (with dimensions of 1cm x 5.5cm each) are controlled individually by digital valves and a rippling effect which resembles that of skin stroking is created when inflating adjacent air cells with an overlapping time [77]. Furthermore, a recent study looking at the properties of the S-CAT found that subjective pleasantness ratings were higher following slow, caress-like affective touch at CT-optimal velocities, as opposed to fast, CT-suboptimal touch, irrespective of stimulation type (i.e., skin-to-skin, brush stroking, or S-CAT touch; [77]). Therefore, this prototype produces psychophysiological effects which can be compared to the CT-optimal affective touch which can be delivered either by a soft brush (like in Experiment 1) or skin-to-skin affective touch [79] when looking at the behavioural data (e.g., [77]; see also S1 File). At the absence, however, of microneurography studies to confirm this and the fact that the S-CAT is still at a prototype level, thus warranting further examination of its ability to optimally stimulate CT-afferents, we refer to the touch delivered by the S-CAT as 'affective touch'. The temperature of the sleeve was kept at 36°C to keep it as close to human body temperatures as possible and reduce potential effects of low temperatures on social intent [89].

## Procedure

**Introduction of participants to the "other users" and the platform.** Upon arrival, participants met other FaceJournal users who were introduced to them in another room at the

onset of the experiment. Participants were told that the other users would be logged into the platform at the same time from that room. Participants were also told that the 'other' users would be able to read the participant's posts that the participant would be writing and posting on their profile in real time, and deciding–also in real time- whether to send them visual, or visuotactile feedback and a number of 'likes'. The participants were told that the visual feedback would be in the form of visual emoticons, and if the "other" users wanted to send tactile feedback as well, they would do that by sending touch via a wearable sleeve (i.e., S-CAT) that the participant would be wearing during the experiment. In reality, the 'other' users (unbeknownst to the participants until the debriefing at the end of the session) were lab members of the research team and they were not involved in the feedback participants received. Instead, the visual and visuotactile feedback the participants would be receiving during the experiment was all predetermined and computer generated. Participants were told they would not be meeting the other users again at any point during the experiment.

**Introduction of participants to the S-CAT.** After meeting the "other" users in a different room, the participants were taken to the room where the experiment would take place and were instructed to sit at a table in front of the laptop they would be using during the task, which was placed behind a makeshift wall so that they could not see the experimenter. Next, the S-CAT was worn on the participant's left forearm and then the experimenter sat in front of a monitor (behind the makeshift wall). Two familiarisation trials with the S-CAT were conducted—the experimenter sent touch via the S-CAT to ensure participants were familiar with the sensation of the mediated touch and that the pressure and intensity of the touch were correct.

**Sending feedback via visual emoticons or visual emoticons with affective touch.** Participants were introduced to the concept of social media (similarly to Experiment 1) and were were first asked to create a profile and then log onto FaceJournal using credentials of their choice (see Fig 3). Once logged in, participants saw "their" post (they were asked to imagine they had written and posted it themselves, on their own platform profile), and then received feedback. After that they were asked to rate the amount of social intent they perceived, before being shown another post (this was repeated for all 48 posts). Each post was first displayed for 6 seconds without any feedback (see Fig 5). Then, for 2 seconds the pre-determined emoticon appeared together with a number of 'likes' which ramped up to the final, pre-determined number of likes associated with each post, starting from 0 (meaning no likes at all). Next, the visual feedback in the form of the visual emoticon was kept on the participant's screen for 12 seconds. Half of the posts received visuotatile feedback, meaning that participants also received touch in addition to the emoticons they saw on their screen for the same overall duration. In those instances, the pre-determined tactile feedback—delivered via the S-CAT—was initiated (i.e., at

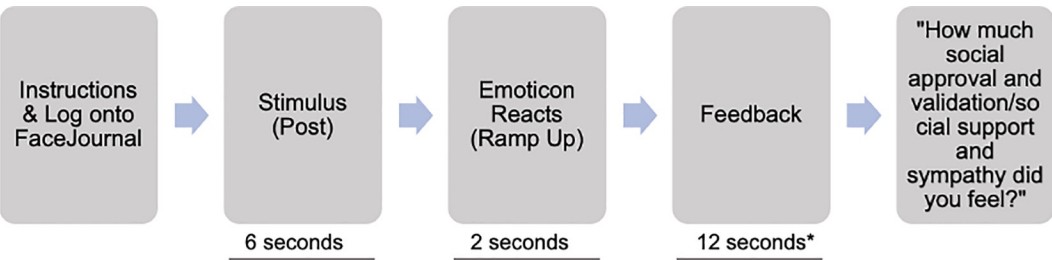

**Fig 5. Trial timeline for Experiment 2.** *Feedback was either the total number of reacts and the visual emoticon displayed on the screen for 12 seconds, or a combination of this aforementioned feedback together with tactile feedback sent to the participant via the wearable sleeve for 12 seconds.

the 9th second until the end of the 20th second). The trial durations here differed to those in Experiment 1, because we observed that participants did not need that long to read the post pre-feedback as they could keep reading the post while receiving feedback. Yet, we gave more time in total to allow them to better imagine these as their own posts, to experience the feedback for longer, and to think of the emotions the other users are trying to convey.

At the end of the task participants completed the self-report questionnaires and some other measures to assess perception of touch, and changes in physiological measures (see more in S1 File). Finally, participants were fully debriefed regarding the computerised and not social nature of the feedback they received, their FaceJournal credentials and profiles were instantly deleted, and a manipulation check question revealed that no participant guessed this deception in advance.

## Results: Experiment 2

### Hypothesis 1: Participants perceived greater social intent when receiving visuotactile feedback on positive posts

As predicted, visuotactile feedback increased perceived social intent significantly more than visual feedback ($\beta = 4.20$, SE = 1.32, $t(3.18)$, 95% CI = 1.61–6.79, $p = 0.001$, ICC = 0.78, $R_{marginal}^2 = 0.011$, $R_{conditional}^2 = 0.781$). We also found a significant interaction between feedback mode and valence ($\beta = 6.27$, SE = 2.61, $t(2.41)$, 95% CI = 1.16–11.38, $p = 0.016$, ICC = 0.77, $R_{marginal}^2 = 0.044$, $R_{conditional}^2 = 0.783$). In planned contrasts exploring the effect of feedback mode in each valence level separately, we found that perceived social intent was rated as significantly greater following visuotactile feedback as opposed to visual feedback on positive posts ($\beta = 7.34$, SE = 1.50, $t(4.89)$, 95% CI = 4.40–10.28, $p<0.001$, ICC = 0.85, $R_{marginal}^2 = 0.033$, $R_{conditional}^2 = 0.856$), but not on negative posts ($\beta = 1.06$, SE = 1.76, $t(0.61)$, 95% CI = -2.39–4.52, $p = 0.545$, ICC = 0.79, $R_{marginal}^2 = 0.001$, $R_{conditional}^2 = 0.791$) (Fig 6).

### Hypothesis 2: Greater levels of perceived social intent were predicted by increased identification with the FaceJournal posts

We also examined whether perceived social intent (DV) following the different types of feedback (visual vs visuotactile; IV) depended on the degree to which participants found the posts relevant (IV) and were able to identify (IV) with them. Due to the high correlation between our two IVs (relevance and identification; $r = 0.65$, $p < .001$), we first performed a PCA to orthogonalise the two IVs and minimise collinearity effects. We then ran two MLMs, one for the main effects (relevance and identification) only, and one for their interaction with perceived social intent and found a significant main effect of identification (($\beta = 4.87$, SE = 1.51, $t(3.23)$, 95% CI = 1.92–7.83, $p = 0.001$, ICC = 0.77, $R_{marginal}^2 = 0.072$, $R_{conditional}^2 = 0.788$), and a non-significant effect of relevance ($\beta = 2.51$, SE = 1.47, $t(1.71)$, 95% CI = -0.37–5.40, $p = 0.088$, ICC = 0.77, $R_{marginal}^2 = 0.072$, $R_{conditional}^2 = 0.788$). That is, the more participants identified with the posts, the greater their reported levels of perceived social intent were. The interactions between identification and feedback mode and relevance and feedback mode were not significant (Feedback Mode x Identification: ($\beta = 1.19$, SE = 1.35, $t(0.88)$, 95% CI = -1.46–3.84, $p = 0.379$, ICC = 0.77, $R_{marginal}^2 = 0.072$, $R_{conditional}^2 = 0.788$); Feedback Mode x Relevance ($\beta = 0.71$, SE = 1.36, $t(0.52)$, 95% CI = -1.95–3.37, $p = 0.600$, ICC = 0.77, $R_{marginal}^2 = 0.072$, $R_{conditional}^2 = 0.788$)). Although not initially preregistered, given our significant effect of identification on perceived social intent, we also ran an analysis to examine whether participants identified more with positive or negative posts but found no significant effect of valence

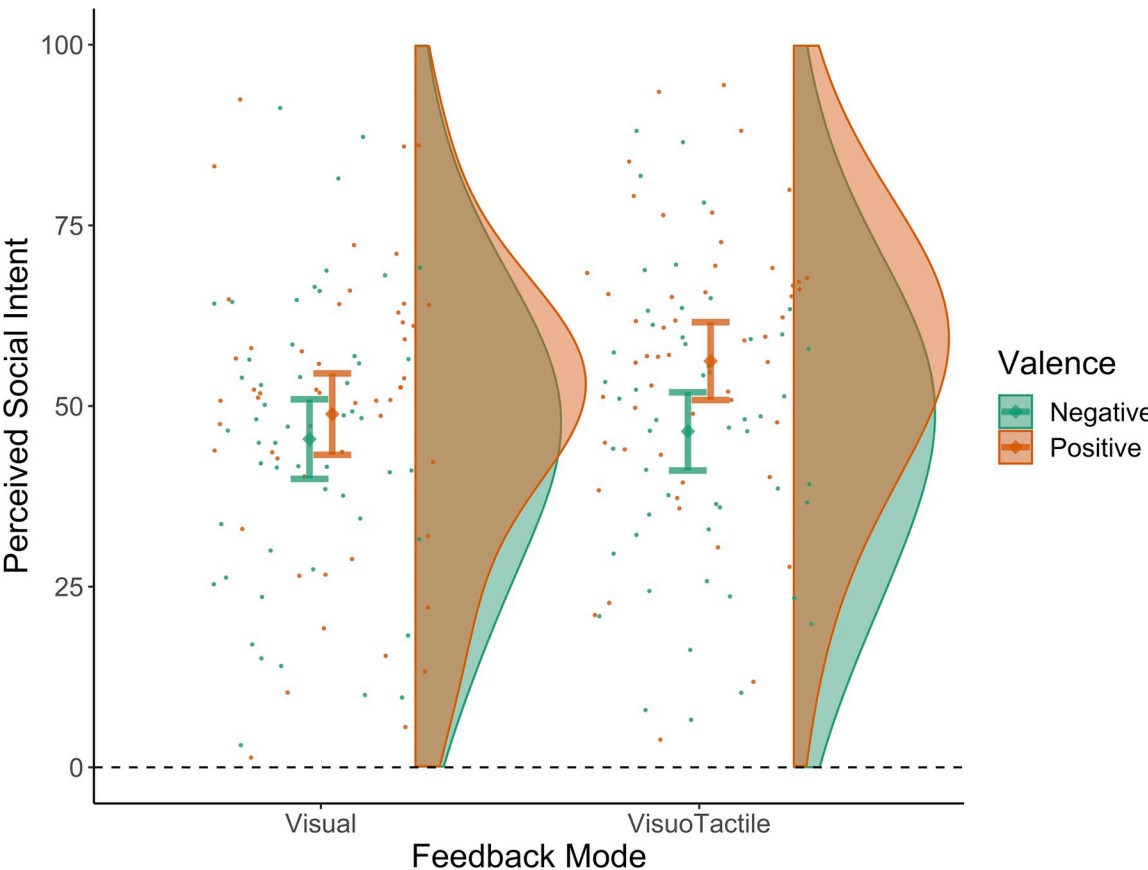

**Fig 6. Social intent scores following visual and visuotactile feedback.** Participants' social intent scores following visual and visuotactile feedback on positive and negative posts. The rainclouds represent the distribution of Perceived Social Intent grouped by Feedback Mode and Valence. The dots represent the average score per participant for each combination of Feedback Mode and Valence. Error bars indicate the 95% confidence interval (CI) around the mean (Mean of Perceived Social Intent +/- 95% CI). The dot on the error bar is the sample mean of Perceived Social Intent for the respective combination of Feedback Mode and Valence. The horizontal dashed line (at y = 0) indicates that the Perceived Social Intent scores could not be below 0.

on identification levels: ($\beta$ = -0.08, SE = 0.14, $t$(-0.59), 95% CI = -0.36–0.19, $p$ = 0.559, $R^2$ = 0.002, $R_{Adj}^2$ = -0.003)).

### Hypothesis 3: Greater levels of perceived social intent were observed in participants who felt safe and trusted the wearable sleeve while receiving touch

We examined whether feelings of Safety and Trust when wearing the S-CAT influenced participants' perception of social intent following visuotactile feedback. We first examined the main effect of Safety, and the interaction between Safety and valence. Next, as preregistered we repeated the interaction analysis within each valence separately. Then we repeated the aforementioned analyses and used Trust as our IV instead of Safety. We found a non-significant trend on the effect of feelings of Safety on perceived social intent ($\beta$ = 0.22, SE = 0.12, $t$(1.87), 95% CI = -0.01–0.46, $p$ = 0.061, ICC = 0.76, $R_{marginal}^2$ = 0.050, $R_{conditional}^2$ = 0.722), but no significant interaction between Safety and valence ($\beta$ = -0.14, SE = 0.09, $t$(-1.52), 95% CI = -0.33–0.04, $p$ = 0.127, ICC = 0.73, $R_{marginal}^2$ = 0.119, $R_{conditional}^2$ = 0.764). When looking at each valence separately, we found no significant main effect of Safety on sentences of positive

valence ($\beta$ = 0.16, SE = 0.13, $t$(1.27), 95% CI = -0.09–0.42, $p$ = 0.205, ICC = 0.06, $R_{marginal}^2$ = 0.030, $R_{conditional}^2$ = 0.087); however, higher perception of Safety led to significantly greater perceived social intent in the sentences of negative valence ($\beta$ = 0.29, SE = 0.12, $t$(2.35), 95% CI = 0.05–0.54, $p$ = 0.019, ICC = 0.06, $R_{marginal}^2$ = 0.0.96, $R_{conditional}^2$ = 0.152). Similarly, we found a significant main effect of Trust: increased trust was predictive of greater perceived social intent ($\beta$ = 0.22, SE = 0.09, $t$(2.34), 95% CI = 0.04–0.41, $p$ = 0.019, ICC = 0.75, $R_{marginal}^2$ = 0.076, $R_{conditional}^2$ = 0.772). Yet, we found no significant interaction between Trust and valence ($\beta$ = 0.07, SE = 0.08, $t$(0.85), 95% CI = -0.09–0.22, $p$ = 0.395, ICC = 0.72, $R_{marginal}^2$ = 0.412, $R_{conditional}^2$ = 0.758). When looking at each valence separately, we found a significant effect of Trust on perceived social intent in positive sentences ($\beta$ = 0.25, SE = 0.10, $t$(2.53), 95% CI = 0.05–0.46, $p$ = 0.015, $R_{adj}^2$ = 0.788; note that the random effects in this analysis did not explain any of the variance, so we conducted a linear regression instead), and a tendency towards significance in the effect of trust in the negative valence ($\beta$ = 0.18, SE = 0.10, $t$(1.76), 95% CI = -0.02–0.38, $p$ = 0.078, ICC = 0.05, $R_{marginal}^2$ = 0.057, $R_{conditional}^2$ = 0.104).

## Hypothesis 4: Visuotactile feedback did not lead to greater regulation of physiological measures in comparison to visual feedback

Contrary to our hypotheses, visuotactile feedback did not regulate physiological measures (namely, heart rate (HR), skin conductance rate (SCR), and heart rate variability (HRV); Fig 7) more than visual feedback (HR: $\beta$ = -0.15, SE = 0.33, $t$(-0.47), 95% CI = -0.79–0.48, $p$ = 0.638, ICC = 0.91, $R_{marginal}^2$ = 0.000, $R_{conditional}^2$ = 0.910; SCR: $\beta$ = -0.02, SE = 0.03, $t$(-0.91), 95% CI = -0.07–0.03, $p$ = 0.365, ICC = 0.89, $R_{marginal}^2$ = 0.001, $R_{conditional}^2$ = 0.887; HRV: $\beta$ = 0.0008, SE = 0.001, $t$(0.769), 95% CI = -0.0012–0.0027, $p$ = 0.442, ICC = 0.93, $R_{marginal}^2$ = 0.000, $R_{conditional}^2$ = 0.927). Interestingly, in preregistered analyses examining the role of rejection sensitivity and attachment anxiety we found that visuotactile feedback led to more downregulation of SCR in individuals with high rejection sensitivity scores in comparison to visual feedback ($\beta$ = 0.01, SE = 0.01, $t$(2.12), 95% CI = 0.00–0.02, $p$ = 0.034, ICC = 0.89, $R_{marginal}^2$ = 0.019, $R_{conditional}^2$ = 0.896). A similar pattern was observed in HRV–both across post valence, and within each valence separately): the regulation in HRV was better explained by the visuotactile feedback in comparison to the visual feedback in individuals with higher trait anxiety scores (Across valences: ($\beta$ = 0.0005, SE = 0.0002, $t$(3.254), 95% CI = 0.0002–0.008, $p$ = 0.001, ICC = 0.93, $R_{marginal}^2$ = 0.064, $R_{conditional}^2$ = 0.933); Negative Valence: ($\beta$ = 0.0005, SE = 0.0002, $t$(2.082), 95% CI = 0.0000–0.001, $p$ = 0.037, ICC = 0.92, $R_{marginal}^2$ = 0.065, $R_{conditional}^2$ = 0.921); for full details see S1 File).

## Summary

In summary, participants reported greater levels of social intent following visuotactile feedback as opposed to visual feedback alone, and especially on posts of positive valence but not when looking at the effects of visuotactile vs. visual feedback on posts of negative valence. Moreover, the extent to which participants identified with (but not related to) the posts was predictive of greater levels of social intent, irrespective of feedback mode. As expected, increased feelings of safety and trust when wearing the S-CAT were predictive of a higher rating of social intent in the visuotactile feedback mode. Moreover, pleasantness ratings were higher following touch delivered at CT-optimal velocities irrespective of how touch was delivered (i.e., via the S-CAT or brush strokes; see S1 File for details on this analysis). Finally, feedback mode was not predictive of physiological (HR, SCR, and HRV) changes, but we found that higher rejection sensitivity scores and higher trait anxiety scores explained more of the SCR and HRV variance, respectively following visuotactile feedback in comparison to visual feedback.

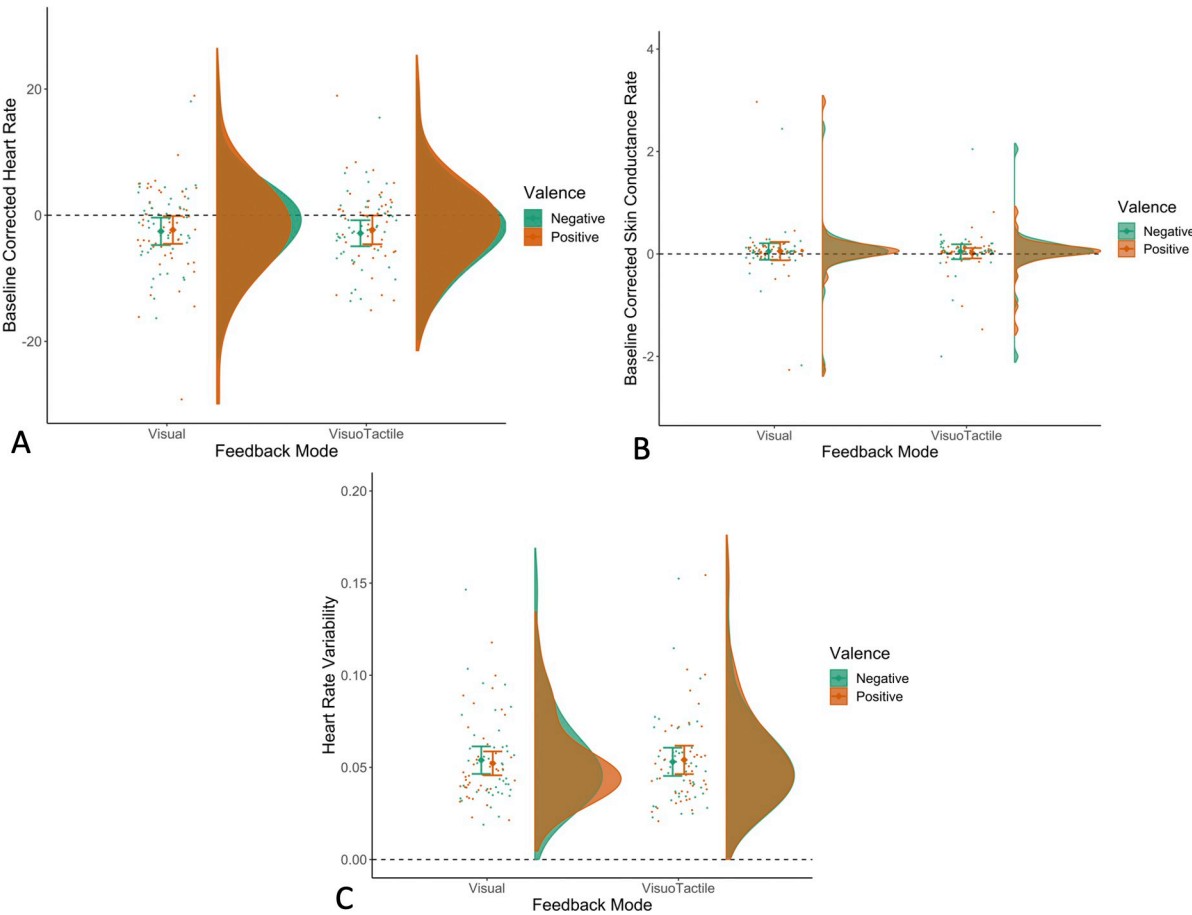

**Fig 7. Changes in physiological measures following visual and visuotactile feedback.** Heart rate (HR; Panel A) and skin conductance rate (SCR; Panel B) were baseline corrected–in other words, the mean HR and SCR scores from each condition block (e.g., visuotactile feedback on negative posts) were subtracted from the average HR and SCR scores we obtained before the task. Heart Rate Variability (HRV; Panel C) was calculated as the time elapsed between successive inter-beat intervals. The y'y axis on Panel C represents the mean of HRV values obtained during the main task, when participants were receiving feedback on their posts. The rainclouds–in all three panels–represent the distribution of the physiological measure grouped by Feedback Mode and Valence. The dots represent the average score per participant for each combination of Feedback Mode and Valence. Error bars indicate the 95% confidence interval (CI) around the mean (Mean of Baseline Corrected HR in Panel A, Baseline Corrected SCR in Panel B, and HRV in Panel C).

## Discussion

Across two experiments we explored the role of affective touch during social media communications. Participants imagined they were posting brief posts on a given social media platform and receiving social feedback. We examined how participants perceived this feedback (referred to as social intent) when it was delivered via standard visual emoticons versus various conditions of tactile feedback, conceptualised as 'tactile emoticons'. In line with our hypothesis, across both Experiments (visuo)tactile feedback was predictive of greater social intent in comparison to visual emoticons, and this effect appeared to be driven by posts of positive valence. Contrary to our predictions, feedback mode did not influence the regulation of physiological measures but there were some indications for further study on exploratory analyses. More generally, we also conducted a set of secondary, exploratory (Experiment 1) and preregistered (Experiment 2) analyses to specify some of our behavioural and physiological findings, control for important confounds and explore individual differences. We discuss all these results and their combined implications below.

Results from both experiments showed that (visuo)tactile feedback was associated with greater levels of perceived social intent compared to visual feedback only, irrespective of whether the feedback was delivered by a previously unfamiliar confederate manually (via a brush) or remotely (mediated via a robotic device). This result is in line with previous studies showing that social touch can convey meaning (e.g., messages of support) without any accompanying verbal communication [11–13], even when touch is given by a stranger [12]. Several haptic communication technologies already exist and preliminary findings have shown support for their benefits, including the ability to enhance social connection between individuals who are physically distanced, imitate a hug and effectively increase feelings of proximity (e.g., [58, 59]). To our knowledge, our study is the first to also show that tactile, affective communication can have added benefits to other frequently used non-verbal communication aids, such as visual emoticons in social media communications. The interpretation of visual emoticons may vary depending the situation and message a sender is trying to convey [73–76] and it appears that prosocial touch could offer such a context and enhance non-verbal communication. Interestingly, perceived social intent was positively correlated both with actual social media engagement frequency and the ability to recognise and describe emotions (as measured using the Toronto Alexithymia Scale (TAS; [90]), and negatively correlated with how aware of and connected one is to what their body communicates (as measured using the Body Awareness Questionnaire; [91]). Interestingly, we found no correlation between attitudes to social touch (as measured using the Social Touch Questionnaire; [92]) and social intent scores. Thus, while we have indications that there can be individual differences in non-verbal emotional communication during social media, future studies with larger samples could further examine how exactly tactile or visuotactile emoticons could alter the habits and the effects of social media use.

Moreover, the fact that tactile feedback delivered via a haptic device was still associated with greater perceived social intent in comparison to only visual feedback adds evidence to the feasibility of developing haptic technologies to enrich existing communications. Yet, Experiment 2 results showed that levels of perceived social intent were greater when participants felt safe and trusted the wearable sleeve. Hence, any future work on robotic devices and mediated-affective touch needs to consider such dimensions of perceived safety and comfort, and the many ethical issues involved in remote touch, such as intrusive or unwanted touch. In this experiment, these aspects were controlled by informed consent and the strict, controlled lab conditions, but when mediated touch is applied outside the lab, and particularly as part of social media interactions with unfamiliar others, such considerations are key. The prosocial effects identified here may be accompanied by unwanted and less positive effects in unregulated interactions. Thus future, applied and field studies should consider such dimensions and other context-specfic effects [93–97].

We also examined how the valence and 'intensity' of the communication influenced the social intent participants felt following (visuo)tactile versus 'visual only' feedback. Specifically, in Experiment 1 participants perceived greater levels of social intent when receiving feedback of high versus low specificity, in both visual and tactile modalities. There are already some relevant studies in the visual domain, such as the finding that individuals will opt for visual emoticons that match the intensity of the emotion one is trying to express [76]. Moreover in the tactile domain, previous studies have shown that stroking touch delivered at CT optimal speeds may convey social intentions differently than non-CT optimal stroking touch (e.g., [27, 33, 48]) and different types of touch seem capable of communicating different emotions, even without any other verbal cues (see Introduction). Our findings further suggest that when considering how to implement tactile emoticons in social media communications, the precise properties of the touch stimulation may also influence how tactile emoticon are perceived. We

only tested the velocity of stroking touch in the present study but similar findings have been observed as regards temperature and vibration in mediated communication [93].

Interestingly, contrary to our predictions, in both experiments we saw that (visuo)tactile feedback enhanced the ratings of social intent in comparison to visual feedback on positive but not negative posts. This effect could be explained by participants identifying more with the positive instead of negative posts but we found that participants' ability to identify with the posts was not influenced by the post's valence. Previous studies have shown that slow, caress-like touch can regulate and soothe negative emotions (e.g., stress, pain of social rejection, physical and social pain [15, 34–41]) and hence this finding was unexpected. However, a few important differences are worth noting. Previous studies mainly measured whether participants' emotional states (e.g., feelings of anxiety or rejection [15]) are regulated via touch, while our study measured the amount of social intent participants perceived upon receiving feedback, not their own mood changes. In terms of the handful of studies that focused on the social emotions and intentions touch can communicate, touch, including CT-optimal stroking touch, was found to be capable to convey sympathy, affiliation/love, lust and social support (see Introduction). While all of these emotions and social intentions are clearly prosocial and in that sense positive, to our knowledge, none of these studies manipulated the valence of the underlying communication as in the present study (but see one study on 'imagined' touch; [98]). For example, while touch may be a useful non-verbal communication cue conveying social sympathy in general, its utility and meaning may be altered if a person is actually upset in the moment, or facing a concurrent threatening situation. Future studies could thus build more specific emotional contexts in which to test the meaning and interpretation of tactile emoticons.

Given that touch has been noted to have positive effects not only mood but also physiology [78–82], we also examined the role of feedback mode on heart rate, SCR and HRV, expecting better regulation of those modalities following visuotactile feedback compared to visual feedback. Contrary to our prediction, visuotactile feedback did not affect participants' physiological measures more than visual feedback. The lack of statistical effects here requires further exploration as it is possible that our physiological measurements were not sensitive enough in this study (and technical errors during testing reduced power), particularly given that single sentences are unlikely to generate important mood and physiological changes that could then be regulated by emoticons. Future studies could implement more prolonged causal designs with positive and negative mood induction and longer periods of measurement to disambiguate between these possibilities. However, it is noteworthy that in further exploratory analyses of the relationship between physiological state regulation and individual differences, as measured using psychometric questionnaires, we found that downregulation of SCR was better explained by the visuotactile feedback as opposed to the visual feedback in individuals with higher rejection sensitivity, and visuotactile feedback caused greater HRV in participants with higher trait anxiety. These results are in line with previous findings showing an association between higher scores in attachment anxiety and reduced pleasantness discrimination between CT-optimal and CT-suboptimal touch [41] and improved cardiovascular reactivity when receiving touch [99]. These findings could therefore form the basis of future studies with targeted recruitment strategies on the role of tactile emoticons in the regulation of the emotions and physiological states that can be created by social media use, particularly in individuals with certain traits.

Our two experiments are the first to implement tactile emoticons in social media communications and explore their role in communicating emotions. Yet, there were several limitations which could be addressed in future studies. One design limitation in Experiment 1 which could have influenced the direction of the results is that the visual emoticons that were used

(e.g., red heart in the high feedback support level conditions) could have been perceived as ambiguous when used together with a post of negative valence (e.g., "I failed"). However, this limitation was directly addressed in Experiment 2. A second limitation is that as regards to social intent scores across the two experiments, we cannot statistically compare to draw conclusions as to why overall participants from both experiments seemed to report increased scores of social intent following (visuo)tactile feedback as opposed to visual feedback only. We speculate that this similarity across the two experiments–at the absence of explicit, quantitative analyses–could be explained by the comparable effects and responses elicited by brush stroke and S-CAT mediated touch. Moreover, our finding that increased feelings of safety and trust towards the S-CAT (in Experiment 2) were predictive of higher social intent scores is in line with another recent study in which we showed that robotic touch effects (e.g., in eliciting calmness) can depend on the degree of perceived pleasantness and awkwardness of the robotic device [100]. A third limitation is that in order to control for semantic, syntactic and emotional content of the alleged interactions participants did not write their own posts which could have impacted their ability to imagine they were receiving feedback on their own posts. While the relevance of the posts to the participants' actual life was not predictive of perceived social intent, we found that as expected, identifying more with the post content (Experiment 2) was predictive of higher levels of perceived social intent and expectations regarding the number of 'likes' participants had for the posts influenced their perceived social intent (Experiment 1). Thus, although we did not find that such factors impacted how social intent was influenced by feedback, in future studies where participants can formulate and post their own content, the perception of social intent may be even stronger for the different types of feedback tested here. Finally, we did not directly compare perception of social intent between manual and mediated affective touch, given the different samples, designs and measurements of Experiments 1 and 2, and thus cannot claim that participants preferred one over the other touch medium for receiving feedback.

## Conclusion

In summary, this study aimed to improve our understanding of the role of touch in social media communications by comparing the effects of (visuo)tactile emoticons, delivered according to certain properties suggested by the neurophysiology of touch, against standard, visual-only emoticons. Irrespective of whether participants received tactile feedback delivered by brush-stroking by a confederate (Experiment 1), or remotely via a wearable sleeve (Experiment 2), tactile and visuotactile emoticons appeared to communicate prosocial intent more clearly, particularly during communications of positively valenced content. Our findings add to the growing body of literature pointing towards the benefits of non-verbal, touch communication. These lab-based results indicate that roboaffective devices could enhance communications that occur during existing social media interactions, however further ethical and emotional dimensions of such communications need to be tested beyond the lab, before the adoption of such tactile emoticons.

## Supporting information

**S1 File.**
(DOCX)

## Acknowledgments

We thank all participants who took part in the experiments. We also thank Prof Sara Price, Dr Frederick Brudy and Dr Douglas Atkinson for their theoretical contributions during

conceptualisation of our pilot studies as well as Faisal Simrah and Can Su Cimenbicer for assistance with data collection.

## Author Contributions

**Conceptualization:** Alkistis Saramandi, Yee Ki Au, Nadia Bianchi-Berthouze, Carey Jewitt, Paul M. Jenkinson, Aikaterini Fotopoulou.

**Data curation:** Alkistis Saramandi, Yee Ki Au, Athanasios Koukoutsakis.

**Formal analysis:** Alkistis Saramandi, Yee Ki Au, Athanasios Koukoutsakis.

**Funding acquisition:** Nadia Bianchi-Berthouze, Carey Jewitt, Aikaterini Fotopoulou.

**Investigation:** Alkistis Saramandi, Yee Ki Au.

**Methodology:** Alkistis Saramandi, Yee Ki Au, Paul M. Jenkinson, Aikaterini Fotopoulou.

**Project administration:** Alkistis Saramandi, Yee Ki Au.

**Resources:** Alkistis Saramandi, Yee Ki Au, Caroline Yan Zheng, Adrian Godwin.

**Software:** Athanasios Koukoutsakis, Caroline Yan Zheng, Adrian Godwin.

**Supervision:** Athanasios Koukoutsakis, Aikaterini Fotopoulou.

**Visualization:** Alkistis Saramandi, Yee Ki Au.

**Writing – original draft:** Alkistis Saramandi, Yee Ki Au.

**Writing – review & editing:** Alkistis Saramandi, Yee Ki Au, Athanasios Koukoutsakis, Paul M. Jenkinson, Aikaterini Fotopoulou.

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
