## [Decision Letter · Decision Letter 0]

15 Dec 2023

PONE-D-23-29955Tactile Emoticons: Conveying Social Emotions and Intentions with Manual and Robotic Tactile Feedback During Social Media CommunicationsPLOS ONE

Dear Dr. Saramandi,

Thank you for submitting your manuscript to PLOS ONE. After careful consideration, we feel that it has merit but does not fully meet PLOS ONE’s publication criteria as it currently stands. Therefore, we invite you to submit a revised version of the manuscript that addresses the points raised during the review process.

We look forward to receiving your revised manuscript.

Kind regards,

Valentina Bruno

Academic Editor

PLOS ONE

Journal Requirements:

Reviewers' comments:

Reviewer's Responses to Questions

**Comments to the Author**

1. Is the manuscript technically sound, and do the data support the conclusions?

Reviewer #1: Yes

Reviewer #2: Yes

2. Has the statistical analysis been performed appropriately and rigorously? 

Reviewer #1: Yes

Reviewer #2: Yes

3. Have the authors made all data underlying the findings in their manuscript fully available?

Reviewer #1: Yes

Reviewer #2: Yes

4. Is the manuscript presented in an intelligible fashion and written in standard English?

Reviewer #1: Yes

Reviewer #2: Yes

5. Review Comments to the Author

Reviewer #1: Overall, I think this is a very nice study that will make a good contribution to the literature. The experiments, analyses and conclusions are all sound. However, I have some substantial concerns with some aspects of the presentation, which leads me to recommend a major revision.

Major comments

The S-CAT appears to be an apparent motion device, with pneumatic elements 1 x 8 cm. The authors report that it delivers a CT optimal stimulus at 6cm/s, analogous to brushing at CT optimal velocities. However, it is far from clear that this device stimulates CT afferents optimally. Wessberg et. al. (2003) report CTs as having a mean receptive field size of 14.7 +/- 10.5 mm sq., so any given CT afferent is likely only stimulated by one S-CAT element. The S-CAT produces apparent motion rather than real motion, and the velocity of the stimulation is derived from the timing between successive activations of pneumatic elements. This means that the individual CTs are not going to be sensitive to the lateral velocity produced by this device.

Wessberg et. al. (2003) and others also showed that CTs respond to indentation. It is likely that the S-CAT does stimulate CTs, but how optimal it is cannot depend on its lateral velocity. I don’t think this issue should worry the authors too much, since it is clearly a pleasant stimulus that is comparable to soft brushing in many ways. However, it is a critical theoretical point that needs to be addressed. It is important to re-frame the nature of this stimulus, perhaps describing it as a “pleasant” or “affective” touch stimulus, rather than CT optimal.

J. Wessberg, H. Olausson, K. W. Fernström, and Å. B. Vallbo, ‘Receptive Field Properties of Unmyelinated Tactile Afferents in the Human Skin’, Journal of Neurophysiology, vol. 89, no. 3, pp. 1567–1575, Mar. 2003, doi: 10.1152/jn.00256.2002.

Expt1 design:

The selection of emoticons for high vs. low support needs to be better motivated. It’s not intuitive to me that thumbs up is high support and blue heart is low support (and why is it called “baseline blue heart”?). How did the authors make the choice? For both tactile and visual feedback, why was habituation not a concern for low support feedback, where only one stimulus was used?

Figure 1: It is certainly useful to have a diagram to explain the fairly complex design, but I did not find this diagram helpful. I noticed the following issues:

- “Valence” and “Feedback mode” are both variable names, but are represented differently, one embedded in the flow chart, the other as an annotation on the side.

- Valence being “positive” or “negative” doesn’t explain anything. Without reading the text, I would wonder what is it the valence of? Some example sentences would be nice here.

- In the text it says there were 8 conditions, but in the diagram it looks like there are four.

- Feedback support level is not represented in the diagram.

- The caption could give more details.

Figures 3 and 7: the caption should describe all the elements of the figure and say what they mean. What does each dot represent, is it a single rating, or an average per participant? What are the confidence intervals, and the point in the middle? Is it mean +/- SD, or something else? EMM and 95% CI? How do these figures relate to the statistical models?

The authors have clearly put a lot of effort into careful consideration of statistical models, power analysis, and pre-registering experiment 2, and I commend them for that. Further, I commend the authors on providing clearly labelled data and reproducible analyses. I did not do a thorough data/code review, but performed a few spot checks, and did not find any issues.

It is strange to me that the SCR and HR results were not included in the main manuscript. It would have been nice to see a figure showing these data. Even though there was no significant main effect, I think the results are of sufficient interest to be included.

Minor comments

The authors have made a thorough review of the relevant literature, which is commendable. I wonder if it is necessary to cite so many review articles, given that in most cases the authors already provide several citations to original research articles.

There are quite a lot of minor typos/spelling/grammatical errors. I started pointing them out, but there were enough that I feel it was getting out of scope for my role as reviewer. I suggest the authors do another editing pass to iron it out.

The IV is sometimes described as “perceived social support” and at other times as “perceived social intent”, and I suggest that a single consistent description should be used. I think the former is more intuitive, especially when reporting results e.g. “the participants perceived greater social support …” reads better than “the participants perceived greater social intent …”

50-55, sentence is very long

90, not clear what is meant by ”experience” here

136-47: somewhere here, briefly describe how the touch is delivered in Experiment 1 (brush)

138 “emtional” should be “emotional”

195 what does “pre-determined” mean here? Was the visuotactile feedback in expt 2 not pre-determined?

202 “resulting in 8 posts were combination” something wrong with this sentence

203-207 why is expt 2 mentioned here when it has its own section later?

Figures 2 and 4 – I’m not sure the content here justifies a whole figure. The visualisation of social media posts could perhaps be incorporated into Figure 1. Alternatively, Figures 2 and 4 could be combined.

Figures 3 and 7 – It is difficult to compare these figures with the valence and feedback format being swapped (colour vs. x-axis). The error bars are very difficult to see, the authors could consider making them more visible in some way, e.g. making the dots partially transparent or paler. The distributions should also be partially transparent since they are overlapping.

482-485 sentence is long and unwieldy

Expt 2 –it is not clear where the participant thinks the touch comes from, do they believe the experimenter chooses when it is delivered, that it is based on emojis/likes etc., or that it is a feedback option that some/all of the “other participants” (confederates) are selecting in addition to emojis?

644-645 – Alpha was set at 0.05, so it is inappropriate to say “We found a tendency towards significance”.

I tried to run the manuscript through http://statcheck.io/, however it didn’t seem to detect most of the tests because of the formatting of the pdf. I recommend the authors use this tool to check for issues in their statistical reporting, e.g. copy/paste errors.

Reviewer #2: This is an interesting, preregistered (exp. 2 only), lab-based study which aims to examine the role of touch in social media communications by comparing the effects of visuo-tactile emoticons, delivered according to CT-optimal and CT-suboptimal properties of affective touch as compared to standard, visual-only emoticons. Authors report that irrespective of whether participants received tactile feedback delivered by brush stroking by a confederate (Exp. 1), or remotely via a wearable sleeve (Exp. 2), tactile and visuo-tactile emoticons appear to overall communicate prosocial intent more clearly, particularly during communications of positively valenced content. The study is very well designed and executed. It also considers interesting individual traits measured by means of several interesting self-reports, which my moderate the effects obtained. I particularly liked the idea of exploring physiological responses associated to (touch mediated) social media communications, despite the negative results obtained according to which HR, HRV and SCR were not downregulated more in the visuo-tactile feedback mode as opposed to the visual feedback mode. Limitations are fairly acknowledged and discussed in the Discussion section. Overall, I strongly support the publication of this study in PLoS One, once, my only minor comments are acknowledged:

Page 10, Line 223: please check not relevant information reported here (date, time?);

Page 20, Line 479: please clarify what you mean by ‘enhanced’ visual feedback;

Page 22, Line 535: please double-check typo ‘penrumatic actuators’ should read pneumatic actuators;

Page 27: it would be good to see a brief summary of the results obtained by experiment 2.

Methods Experiment 1 (and 2): it would be good to see additional information about ethnicity of participants, and/or in general a more thorough description of the demographic of the two populations (also to allow for better comparison between the two populations of the two studies).

Discussion

Whilst I appreciate a direct comparison between perception of social intent between manual and mediated affective touch, given the different samples, designs and measurements of Experiments 1 and 2, cannot be drawn, yet, it would be useful to provide an explanation (or speculation) as to why overall, differences do not occur in the two touch media. I was actually surprised (apologies if I missed this!) by the fact that I do not seem to find a self-report evaluation of the preference (pleasantness) or intensity for touch induced manually as opposed to touch delivered by the soft robotic device for receiving feedback.

It would be good to see a bit more consideration of the secondary results obtained by using self-report scales which go beyond correlations with the TAS, particularly in relation to the STQ and BAS, and beyond whether individual differences in these traits can be linked to the amount of perceived social intent during the main task as reported in the supplemental analyses (page 29, Lines: 696-699).

6. PLOS authors have the option to publish the peer review history of their article (what does this mean?). If published, this will include your full peer review and any attached files.

Reviewer #1: **Yes: **Sarah McIntyre

Reviewer #2: No

---

## [Author Response · Author response to Decision Letter 0]

30 Apr 2024

We thank the reviewers and the editor for taking time to make insightful suggestions and provide detailed feedback to improve our manuscript. We have uploaded our response to reviewers as well as the revised manuscript and supplementary material.

---

## [Decision Letter · Decision Letter 1]

13 May 2024

Tactile Emoticons: Conveying Social Emotions and Intentions with Manual and Robotic Tactile Feedback During Social Media Communications

PONE-D-23-29955R1

Dear Dr. Saramandi,

We’re pleased to inform you that your manuscript has been judged scientifically suitable for publication and will be formally accepted for publication once it meets all outstanding technical requirements.

Kind regards,

Valentina Bruno

Academic Editor

PLOS ONE

Additional Editor Comments (optional):

Reviewers' comments:

Reviewer's Responses to Questions

**Comments to the Author**

1. If the authors have adequately addressed your comments raised in a previous round of review and you feel that this manuscript is now acceptable for publication, you may indicate that here to bypass the “Comments to the Author” section, enter your conflict of interest statement in the “Confidential to Editor” section, and submit your "Accept" recommendation.

Reviewer #1: All comments have been addressed

Reviewer #2: All comments have been addressed

2. Is the manuscript technically sound, and do the data support the conclusions?

Reviewer #1: Yes

Reviewer #2: Yes

3. Has the statistical analysis been performed appropriately and rigorously? 

Reviewer #1: Yes

Reviewer #2: Yes

4. Have the authors made all data underlying the findings in their manuscript fully available?

Reviewer #1: Yes

Reviewer #2: Yes

5. Is the manuscript presented in an intelligible fashion and written in standard English?

Reviewer #1: Yes

Reviewer #2: Yes

6. Review Comments to the Author

Reviewer #1: I am satisfied that the authors have adressed all my comments. I particularly like the new Figures 1 and 2 which convey the experimental designs clearly at a glance.

Purely as a note of interest, I wanted to respond to the following text that the authors added:

"At the absence, however, of microneurography studies to confirm this and the fact that the SCAT is still at a prototype level, thus warranting further examination of its ability to optimally stimulate CT-afferents, we refer to the touch delivered by the S-CAT as ‘affective touch’."

I would not necessarily require that microneurography data be collected with the S-CAT specifically. Microneurography data with a similar type of device or stimulation would be sufficient to infer how the S-CAT is likely to stimulate CTs with its different speeds and settings. The truth is there are lots of stimulus types for which CT responses are not well characterised, in contrast to A-beta touch afferents.

Reviewer #2: I would like to congratulate the authors on the excellent review work done on their manuscript. I therefore recommend accepting the manuscript for publication in Plos One.

7. PLOS authors have the option to publish the peer review history of their article (what does this mean?). If published, this will include your full peer review and any attached files.

Reviewer #1: **Yes: **Sarah McIntyre

Reviewer #2: No

---

## [Editor Report · Acceptance letter]

20 May 2024

PONE-D-23-29955R1 

PLOS ONE

Dear Dr. Saramandi, 

I'm pleased to inform you that your manuscript has been deemed suitable for publication in PLOS ONE. Congratulations! Your manuscript is now being handed over to our production team.

Kind regards, 

on behalf of

Dr. Valentina Bruno 

Academic Editor

PLOS ONE